# Blood Growth Factor Levels in Patients with Systemic Lupus Erythematosus: High Neuregulin-1 Is Associated with Comorbid Cardiovascular Pathology

**DOI:** 10.3390/life14101305

**Published:** 2024-10-14

**Authors:** Evgeny A. Ermakov, Mark M. Melamud, Anastasiia S. Boiko, Svetlana A. Ivanova, Alexey E. Sizikov, Georgy A. Nevinsky, Valentina N. Buneva

**Affiliations:** 1Institute of Chemical Biology and Fundamental Medicine, Siberian Branch of the Russian Academy of Sciences, 630090 Novosibirsk, Russia; marken94@mail.ru (M.M.M.); alex.sizikov.as@gmail.com (A.E.S.); nevinsky@niboch.nsc.ru (G.A.N.); 2Department of Natural Sciences, Novosibirsk State University, 630090 Novosibirsk, Russia; 3Mental Health Research Institute, Tomsk National Research Medical Center of the Russian Academy of Sciences, 634014 Tomsk, Russia; anastasya-iv@yandex.ru; 4Psychiatry, Addictology and Psychotherapy Department, Siberian State Medical University, 634050 Tomsk, Russia; ivanovaniipz@gmail.com; 5Institute of Clinical Immunology, Siberian Branch of the Russian Academy of Sciences, 630099 Novosibirsk, Russia

**Keywords:** systemic lupus erythematosus, SLE, cardiovascular diseases, serum, neuregulin-1, NRG1, GM-CSF, GDNF, NGF

## Abstract

Patients with systemic lupus erythematosus (SLE) are known to frequently suffer from comorbid cardiovascular diseases (CVDs). There are abundant data on cytokine levels and their role in the pathogenesis of SLE, while growth factors have received much less attention. The aim of this study was to analyze growth factor levels in SLE patients and their association with the presence of comorbid CVDs. The serum concentrations for the granulocyte-macrophage colony-stimulating factor (GM-CSF), nerve growth factor β (NGFβ), glial cell line-derived neurotrophic factor (GDNF), and neuregulin-1 β (NRG-1β) were determined in the SLE patients (n = 35) and healthy individuals (n = 38) by a Luminex multiplex assay. The NGFβ and NRG-1β concentrations were shown to be significantly higher in the total group of SLE patients (median [Q1–Q3]: 3.6 [1.3–4.5] and 52.5 [8.5–148], respectively) compared with the healthy individuals (2.9 [1.3–3.4] and 13.7 [4.4–42] ng/mL, respectively). The GM-CSF and GDNF levels did not differ. Interestingly, elevated NRG-1β levels were associated with the presence of CVDs, as SLE patients with CVDs had significantly higher NRG-1β levels (99 [22–242]) compared with the controls (13.7 [4.4–42]) and patients without CVDs (19 [9–80] ng/mL). The model for the binary classification of SLE patients with and without CVDs based on the NRG-1β level had an average predictive ability (AUC = 0.67). Thus, altered levels of growth factors may be associated with comorbid CVDs in SLE patients.

## 1. Introduction

Systemic lupus erythematosus (SLE) is a severe autoimmune disease that dramatically affects the life quality of patients. The prevalence of SLE ranges from 13 to 7713 persons per 100,000 individuals [1]. Regardless of ethnic groups, women suffer from this disease much more often than men [2]. SLE most often debuts in women at the reproductive age of 25–39 years [3]. The social significance of this disease necessitates the study of the pathogenetic mechanisms of SLE and the development of new methods of diagnosis and treatment.

The pathogenesis of SLE is associated with systemic inflammation, primarily affecting connective tissue [4]. SLE is accompanied by the formation of a wide range of autoantibodies, including anti-nuclear antibodies, forming immune complexes and causing tissue damage [5]. SLE is also associated with an excessive formation of neutrophil extracellular traps (NETs), an impaired clearance of nuclear components [6], a repolarization of macrophages into an M1 phenotype [7], and T-cell stimulation of an autoantibody formation by the B cells in the lymphoid organs and in the kidney [8] and others.

SLE is often accompanied by the development of multiple comorbidities. Cardiovascular disease (CVD) is one of the most common comorbid conditions in SLE [9]. Ischemic heart disease, stroke, and atherosclerosis are significantly more prevalent in SLE patients than in the general population [10,11,12]. Arterial hypertension is also common in SLE patients, especially among women, with some reports indicating up to 74% [13,14]. Both traditional risk factors and disease-related factors play an important role in the development of CVDs in SLE patients [9,15]. There is evidence that traditional CVD risk factors such as smoking, dyslipidemia, metabolic syndrome, and others are more prevalent in SLE patients than in the general population [9]. Lupus nephritis and nephrotic syndrome, which often accompany SLE, are also risk factors for CVD [16,17]. SLE patients with lupus nephritis have a greater risk of hyperlipidemia, which in turn is associated with nephrotic syndrome [16]. A prolonged disease duration and antiphospholipid syndrome are also considered risk factors for cardiovascular complications in SLE patients [18]. In addition, the immunological mechanisms associated with SLE such as inflammation, cytokine dysregulation, and autoantibodies also contribute to the increased risk of CVDs [19]. Some drugs for the treatment of SLE, especially glucocorticoids, may also be associated with the development of CVDs [9,20]. Thus, the pathogenic mechanisms of SLE and CVDs are overlapping and inextricably linked. Interestingly, a recent Mendelian randomization study confirmed the causal effect of SLE on the etiology of CVDs [21]. Cardiovascular complications are one of the leading causes of death among patients with SLE [22]. However, the biomarkers of comorbid CVDs in SLE are still poorly understood.

Cytokine dysregulation is considered to be actively involved in the pathogenesis of SLE. Currently, there are many meta-analyses confirming increased levels of pro-inflammatory cytokines in SLE patients [23,24,25,26]. However, the growth factors that may also be involved in the pathogenesis of SLE are poorly understood compared to cytokines. The granulocyte-macrophage colony-stimulating factor (GM-CSF) is one of the growth factors associated with the pathogenesis of SLE. The GM-CSF has pleiotropic effects and is involved not only in the regulation of hematopoiesis but also in the function of many immune cells [27]. The GM-CSF reduces neutrophil apoptosis in juvenile SLE patients [28]. GM-CSF-deficient mice are characterized by a SLE-like phenotype [29]. However, there are few data on the GM-CSF concentration in the blood of SLE patients [30].

Neurotrophic factors are even less studied than hematopoietic growth factors in SLE patients, although the nervous system is also affected in this disease. The neuropsychiatric systemic lupus erythematosus (NPSLE) manifests in about 17.6–44.5% of patients [31]. Some neurotrophic factors, in addition to maintaining nerve cells, are also involved in the regulation of immune cells. For example, the nerve growth factor β (NGFβ) affects B cells and promotes their differentiation into immunoglobulin-secreting plasma cells [32]. The glial cell line-derived neurotrophic factor (GDNF) has mostly anti-inflammatory functions as it inhibits the synthesis of inflammatory cytokines such as the tumor necrosis factor α (TNF-α), interleukin (IL)-1β, IL-6, and IL-8 by immune cells through the suppression of IL-17 [33]. Other neurotrophic factors and neuregulins, including neuregulin-1 β (NRG-1β), are not only associated with the regulation of the nervous and immune systems but also are involved in the maintenance of the heart and cardiovascular system [34]. However, the concentration of growth factors in SLE patients with comorbid CVDs is also poorly investigated.

In this study, the concentrations of GM-CSF, NGFβ, GDNF, and NRG-1β were investigated in patients with SLE depending on the presence of comorbid CVDs.

## 2. Materials and Methods

### 2.1. Study Subjects

In this case-control study, the serum levels of growth factors were first compared in a group of healthy individuals (n = 38) and the total group of SLE patients (n = 35) and then in the subgroups of patients with (n = 17) and without concomitant CVDs (n = 18). The recruitment of SLE patients and healthy individuals was organized at the Institute of Clinical Immunology (Novosibirsk, Russia). 

The inclusion criteria for the patient group were as follows: confirmed diagnosis of SLE (M32, ICD-10), and signed consent to participate in this study. The diagnosis of SLE was made in accordance with national (Russian Association of Rheumatologists) and international (EULAR [35]) recommendations based on the laboratory and physical tests described previously [36]. Anti-DNA antibody levels were assessed by ELISA using the Vecto-dsDNA-IgG kit (catalog # 8656, Vector-Best, Novosibirsk, Russia). The SELENA-SLEDAI scale was used to assess SLE disease activity [37]. Healthy individuals with no history of autoimmune, cardiovascular pathology, or other somatic diseases living in the same region as the patients were included in this study as a control group.

The exclusion criteria for patients were as follows: the history of a concomitant autoimmune and oncological disease, terminal stages of cardiac and renal failure, lupus nephritis, neuropsychiatric SLE, traumatic brain injury, concomitant neurodegenerative diseases, drug and alcohol addiction, diabetes, and pregnancy. Patients with a history of allergic reactions and infections in the last month were also excluded from this study.

Among the SLE patients, the presence of cardiovascular comorbidities was carefully monitored. The total group of SLE patients were divided into two subgroups according to the presence of CVDs. The presence of a CVD was diagnosed by experienced physicians in accordance with national (Russian Society of Cardiology) and international (The European Society of Cardiology) recommendations [38,39]. The subgroup of patients with CVDs included patients with arterial hypertension, congestive heart failure (CHF), arrhythmias, cardiomyopathy, angina pectoris, vascular pathologies (atherosclerosis, ischemic conditions), and others. Forming subgroups of patients with specific CVDs was not possible due to the small sample size. Individualized data on concomitant CVDs for the SLE patients included in this study are presented in Appendix A.

The SLE patients were taking therapies affecting their immune systems. Each patient received at least two immunotropic drugs, one of which belonged to the glucocorticoid class (dexamethasone, prednisolone, methylprednisolone, betamethasone). Other immunotropic drugs included methotrexate, hydroxychloroquine, azathioprine, filgrastim, mycophenolate mofetil, tenoxicam, and celecoxib. The patients did not receive genetically engineered biological drugs (monoclonal antibodies). In addition, some patients were receiving medications to treat CVDs (antihypertensives, beta-adrenoblockers, and others).

### 2.2. Serum Samples Collection

Blood samples from each participant were obtained by venipuncture in a BD Vacutainer tube with a clot activator (BD, Franklin Lakes, NJ, USA). The tubes were then centrifuged at 2000 g at 4 °C for 30 min. The resulting plasma samples were aliquoted and stored at −80 °C until the analysis.

### 2.3. Multiplex Analysis of Growth Factor Levels in Serum

The concentration of the growth factors in the serum was determined by the Magnetic Luminex assay using a custom Human Premixed Multi-Analyte Kit (catalog # LXSAHM, R&D Systems, Minneapolis, MN, USA), following the manufacturer’s protocol. The concentrations of the following growth factors were analyzed: GM-CSF, GDNF, NGFβ, and NRG-1β. However, NRG-1 is known to have six types and many alternatively spliced isoforms of the *NRG1* gene [40]. Hence, more specifically the NRG-1β type 1 (hereinafter NRG-1β for brevity), i.e., one of the functionally active isoforms containing a soluble epidermal growth factor (EGF)-like domain, was analyzed using this kit according to the manufacturer’s description. All measurements were carried out on a MagPix multiplex analyzer and Luminex 200 Instrument System (located in the Core Facility “Medical genomics”, Tomsk NRMC). The obtained signals were processed in the Luminex xPONENT software and exported to the MILLIPLEX Analyst 5.1 program for a final analysis. The results were expressed in pg/mL.

### 2.4. Protein–Protein Association Network Analysis

The STRING 12.0 online tool (https://string-db.org/cgi/input, accessed on 20 August 2024) was used to construct the protein–protein association networks of the analyzed growth factors with cytokines [41]. The functional interactions of the growth factors and some cytokines whose levels correlated with the growth factor levels were analyzed using this tool (as in [42]). The medium confidence interaction score (0.4) was used for the STRING analysis. The co-expression of growth factors and cytokines was analyzed using the co-expression module of the STRING 12.0 online platform. The co-expression scores were based on RNA expression patterns and on protein co-regulation data provided by ProteomeHD [43].

### 2.5. Statistical Analysis

The statistical analysis of the obtained data was carried out in the STATISTICA 10 (StatSoft, Tulsa, OK, USA) and OriginPro 2021 (OriginLab, Northampton, MA, USA). The Shapiro–Wilk test was used to assess the type of distribution (conformity to normal distribution). Since most of the data had a non-normal distribution, non-parametric criteria were used. The categorical variables were analyzed using the Pearson’s chi-squared test. The Mann–Whitney test was used to analyze the significance of the differences between the two groups, including the growth factor concentrations. In addition, the Kruskal–Wallis test with Dunn’s post hoc test for multiple comparisons was applied to analyze the significance of the differences between the multiple samples. A receiver operating characteristics (ROC) analysis was used to evaluate the quality of the binary classification based on the growth factor concentrations (implemented in OriginPro 2021). Logistic regression was used to build a model to predict the presence of CVD by the NRG-1β level (implemented in STATISTICA 10). Multiple linear regression was used to assess the effect of the level of growth factors on the severity of SLE according to the SELENA-SLEDAI scale (implemented in STATISTICA 10). Spearman’s rank correlation was used to analyze the correlation of the growth factor levels with the cytokines and clinical data. Data on the cytokine concentrations were obtained from our previously published work in which the samples of participants overlapped significantly with the sample of this study [44]. The outliers represented by the dots on the plots were identified using Tukey’s test. OriginPro 2021 and Microsoft Excel 2021 (Microsoft Corporation, Redmond, WA, USA) were utilized to graphically present the data.

## 3. Results

### 3.1. Clinical Characteristics of the Analyzed Groups and the Structure of Comorbid CVDs in Patients with SLE

Thirty-five individuals suffering from SLE were included in the patient group. Depending on the presence of CVDs, this group was divided into two subgroups: with (n = 17) and without CVDs (n = 18). Thirty-eight individuals were included in the group of healthy subjects (HSs group). The clinical characteristics of the study groups are presented in Table 1. The analyzed groups did not differ in age, sex ratio, proportion of smokers, and body mass index. Most of the SLE patients were in the active stage of the disease. The SLE patients with comorbid CVDs had a slightly longer disease duration, although there were no significant differences compared to the patients without CVDs. The atherogenic index of plasma did not differ between the subgroups of patients. The anti-DNA antibody levels were significantly elevated in the total group of SLE patients, although there were no differences between the patients with and without CVDs (Table 1). Each SLE patient was taking therapies affecting the immune system (for details see Section 2.1). Information on the percentage of SLE patients with and without CVDs who received a particular immunosuppressive drug and the proportions of patients receiving from one to five drugs are presented in Appendix A. The percentage of patients taking a specific immunosuppressive drug did not differ between the patients with and without CVDs. However, the majority of patients without a CVD (56.3%) were taking two immunosuppressive drugs, while 53% of patients with a CVD were taking three drugs (Appendix A). 

Approximately half of the SLE patients had various comorbid CVDs. The structure of the comorbid CVDs in the SLE patients is presented in Figure 1. About one-third of the patients had only one CVD, 41% of the patients had two CVDs, and 24% suffered from three or more cardiovascular pathologies (Figure 1A). Among the defined CVDs, arterial hypertension was the most common, in particular 12 patients out of 17 suffered from this comorbid pathology (Figure 1B). CHF occurred in 22% of patients. Other comorbid conditions such as atherosclerosis, arrhythmias, cardiomyopathies, angina pectoris, and others were even less common. Data on the specific CVDs for each SLE patient are presented in Appendix A.

Information on comorbid pathology (except for CVD) in the groups of patients with and without CVDs is presented in Appendix A. It was shown that osteoporosis, dorsopathies, neuropathies, and chronic tonsillitis (in remission) occurred in the patients of both groups. However, there were no significant differences in the frequency of these pathologies in the groups of patients with and without CVDs.

### 3.2. Serum Concentrations of Growth Factors in Healthy Subjects and SLE Patients

The results of the multiplex analysis of the growth factor concentrations in the total group of SLE patients compared to the group of healthy individuals are presented in Figure 2. The serum concentration of NGFβ was significantly higher (1.2 times) in the SLE patients (median (Q1–Q3); mean ± SD: 3.6 (1.3–4.5); 5.6 ± 5.6) compared to the controls (2.9 (1.3–3.4); 4.1 ± 4.0 ng/mL). The concentration of NRG-1β was also 3.9 times higher in the SLE patients (52.5 (8.5–148); 141 ± 203) than in the healthy subjects (13.7 (4.4–42); 44±68 ng/mL). The GM-CSF levels did not change significantly (HSs: 1.8 (0.5–3.0); 2.8 ± 2.6 vs. SLE: 2.2 (0.2–2.7); 2.7 ± 3.3 ng/mL). There were also no significant differences in the GDNF levels (HSs: 1.9 (0.9–2.4); 1.9 ± 0.8 vs. SLE: 1.5 (0.5–2.4); 2.2 ± 2.9 ng/mL). The samples had some outliers (identified by Tukey’s test) (Figure 2). However, after removing the outliers, the results did not change.

The ROC analysis (Figure 2E) showed that the binary classification models of the healthy individuals and SLE patients based on the NGFβ and NRG-1β concentrations had an average predictive ability. The resulting models were statistically significant (*p* < 0.01). The area under curve (AUC) was approximately the same for NGFβ and NRG-1β.

### 3.3. Clinical Associations

Comorbid conditions, especially CVDs, could be associated with changes in the level of growth factors in SLE patients. Therefore, the levels of growth factors in the subgroups of SLE patients with and without CVDs compared with the healthy subjects were investigated (Table 2). It was shown that the levels of NGFβ, GM-CSF, and GDNF in the two subgroups of SLE patients were not significantly different and corresponded to the levels in the healthy individuals (Table 2). Therefore, the presence of CVDs had no effect on the levels of these growth factors.

Interestingly, the concentration of NRG-1β was found to be higher in the SLE patients with CVDs compared to the healthy individuals and patients without CVDs (Table 2, Figure 3A). There was a trend toward increased NRG-1β levels in the SLE patients without CVDs compared with the controls, but the differences were not significant (*p* = 0.15). Thus, the increase in the NRG-1β levels in the total group of SLE patients (please see Figure 2C) is mainly associated with the contribution of the CVDs.

The ROC analysis showed that the model for the binary classification of the SLE patients with and without CVDs based on the NRG-1β level had an average predictive ability (Figure 3B). In addition, no patterns in the NRG-1β levels in the SLE patients depending on the specific CVD were identified (Appendix A).

A logistic regression model using NRG-1β as the independent variable and the presence of a CVD (binary variable) as the dependent variable was also constructed. However, the developed model was not statistically significant (final loss: 23.36, Chi^2^(1) = 1.765, *p* = 0.18). Therefore, the serum NRG-1β concentration weakly predicted the presence of CVDs in the SLE patients.

The growth factor levels in the healthy individuals and patients depending on their age (less than 40 and more than 40 years) were also analyzed (Appendix A). It was shown that the levels of NGFβ, GM-CSF, and NRG-1β did not differ depending on age. However, the GDNF levels increased with age in the healthy individuals, in particular, the GDNF levels were higher in people after 40 years of age than before 40 years of age. No such pattern was found for the SLE patients (Appendix A). It has also been shown that the levels of growth factors in the SLE patients did not differ depending on disease duration (less than 10 years and more than 10 years, see Appendix A) and disease severity (subgroups according to SELENA-SLEDAI score: Low, 3–5; Medium, 6–10; High, >11; see Appendix A).

The results of the multiple linear regression (Appendix A) showed that none of the independent variables (levels of growth factors) were significant predictors of SELENA-SLEDAI score (as the dependent variable).

### 3.4. Correlation of Growth Factor Levels with Cytokines, Functional Connectivity, and Co-Expression

The sample of patients and healthy individuals overlapped partially with the sample used previously for a cytokine analysis in our previous study [44]. This provided an opportunity to analyze the correlation of the growth factor levels with the number of cytokines in the blood of the participants. The results of the correlation analysis of the growth factor levels with the clinical data and cytokine levels are presented in Figure 4. The exact values of the correlation coefficients and *p*-values are presented in Appendix A.

No significant correlations with the clinical data were found, except for a negative correlation of the NRG-1β levels with the age of the SLE patients without CVDs (Figure 4C) and a positive correlation of the NRG-1β levels with disease duration in the SLE patients with CVDs (Figure 4D).

Growth factor levels have also been shown to positively correlate with the concentration of a number of pro-inflammatory cytokines (Figure 4). Healthy individuals were characterized by the presence of a large number of correlations, some of which persisted in the SLE patients. The most prominent was the correlation of NRG-1β with IL-21, which was high and significant in all groups. NRG-1β also correlated well with the interferon α (IFNα), except for in the SLE patients with CVDs. NGFβ correlated with IL-4 and IFNα, except for in the SLE patients without CVDs. GM-CSF correlated with GDNF, IL-1β, IL-2, IL-4, and IL-21. GDNF also correlated with many cytokines, including IL-1β, IL-2, IL-4, IL-21, and others. Some correlations were lost in patients, for example, IL-6 correlated with all the growth factors in the healthy individuals, but this correlation was lost in all the patient groups. Interestingly, growth factor levels did not correlate with the anti-inflammatory cytokine IL-10 in any of the groups. The growth factors also correlated with each other in the healthy individuals, but in the patients these correlations were lost, only the correlation of GM-CSF with GDNF remained (Figure 4).

The cytokines and growth factors correlating with each other were selected for a protein–protein and functional interaction analysis and co-expression analysis (Figure 5). The cytokines and growth factors were shown to have functional and protein–protein interactions (Figure 5A). The cytokines formed a large network and were functionally associated with each other. IL-1β, IL-2, and IL-4 had marked functional relationships. The growth factors (GDNF, NGFβ, NRG-1β, and GM-CSF) were functionally interconnected with each other. NGFβ, GDNF, and especially GM-CSF were also associated with cytokines. NRG-1β was functionally associated only with growth factors (GDNF and NGFβ) but not with cytokines.

Interestingly, a number of significant correlations persisting in the healthy individuals and SLE patients (for example, correlations of GM-CSF with IL-1β and IL-2) (Figure 4) were confirmed by the protein–protein and co-expression data (Figure 5). However, the marked correlation of NRG-1β with IL-21, GM-CSF with GDNF, and GDNF with IL-2 and IL-4 was not confirmed by the functional interaction and co-expression data. It is conceivable that these molecules may be functionally connected but this requires confirmation.

## 4. Discussion

Growth factors may be involved in the pathogenesis of SLE but are less well studied than cytokines. This paper showed increased serum concentrations of NGFβ and NRG-1β in the SLE patients compared to the healthy individuals, while the GM-CSF and GDNF levels did not differ (Figure 2). Interestingly, the increased NRG-1β levels were associated with the presence of CVDs (Table 2, Figure 3). At the same time, NGFβ was not associated with comorbid CVDs. To the best of our knowledge, this is the first study to investigate the NRG-1β levels in SLE patients. Previous studies have indicated the important role of NRG-1β in cardiomyocyte development and the maintenance of cardiovascular functions in norm and pathology [45,46]. Therefore, the increased NRG-1β levels in the SLE patients with CVDs may be associated with a compensatory response to cardiovascular damage in this disease. NRG-1β can also be considered a potential biomarker of CVDs in SLE patients. However, this study showed an average predictive ability for NRG-1β in the classification of patients with and without CVDs (Figure 3), which may be related to the heterogeneity of CVDs in patients. It is likely that different CVDs may affect NRG-1β levels differently. However, the sample size of this study did not allow us to examine individual CVDs. Therefore, further studies may be aimed at elucidating the contribution of specific CVDs to NRG-1β levels in SLE patients.

The findings of increased NGFβ in SLE patients obtained in this study are consistent with the literature. The blood NGFβ levels were elevated in both the adult and pediatric SLE patients and correlated with disease severity [47,48,49]. However, this study did not show a correlation of NGFβ levels with disease severity and other clinical parameters. NGF has traditionally been considered a neurotrophic factor regulating innervation and the neuronal activity of peripheral neurons [50]. However, NGF has also been shown to be involved in the regulation of immune cell function and inflammation [32]. For example, NGF promotes the proliferation and differentiation of B-cells into plasma cells and the secretion of immunoglobulins [51]. In addition, NGF regulates neutrophil survival by inhibiting apoptosis [52] while neutrophils are known to actively die via NETosis in SLE patients [6]. Therefore, the increase in NGFβ revealed in this study reflects the general pro-inflammatory state of SLE patients and the possible involvement of NGFβ in pathogenetic processes. However, since no association of the NGFβ levels with comorbid CVDs was found, it can be assumed that NGFβ is not involved in cardiovascular injury in SLE patients.

According to the results of this study, the levels of GDNF and GM-CSF were not altered in the SLE patients. The literature data on the concentration of these growth factors are scarce. It is known that in addition to its neuroprotective properties [53], GDNF is known to be an important factor regulating nephrogenesis and supporting renal function [54]. Elevated GDNF levels have been found in lupus nephritis but not in SLE without renal complications [55]. The lack of differences in the GDNF levels shown in this study can be explained by the absence of patients with lupus nephritis in the sample as they were excluded from this study.

There are few data on the concentration of GM-CSF in the blood of patients with SLE. The studies on juvenile SLE have revealed a decrease in GM-CSF concentrations [56]. The protective role of GM-CSF in juvenile SLE patients has also been shown [28]. However, in adult patients, no changes in GM-CSF concentrations have been reported [30]. Our recent study also showed no significant differences in the GM-CSF levels in the SLE patients compared to the controls [42]. Thus, the present data indicate that there is no association of GM-CSF levels with SLE at least in adult patients.

In this study, some correlations of the growth factor levels with the clinical parameters were also found. The NRG-1β concentrations decreased with age in the SLE patients without CVDs (Figure 4C). There is evidence that NRG-1 is associated with longevity (in animal models) [57], so it can be hypothesized that its decrease may lead to accelerated aging in SLE patients. In addition, a positive correlation of NRG-1β with disease duration in SLE patients with CVDs has been shown (Figure 4D), further supporting the association of NRG-1β with cardiovascular pathology.

The data on the correlation of the growth factor levels with the cytokines (Figure 4 and Figure 5) confirmed the existing data on their functional associations and co-expression (for example, GM-CSF with IL-1β and IL-2). However, the obtained data may also indicate the existence of previously unknown functional associations of NRG-1β with IL-21, GM-CSF with GDNF, and GDNF with IL-2 and IL-4. Such associations have not yet been described in the literature; there is only a reference to the fact that GM-CSF administration promotes the up-regulation of GDNF in eye tissues in a rat model [58], indirectly confirming the association of GM-CSF with GDNF. Thus, the results of this study provide directions for future research, especially in examining the functional relationships of the growth factors and cytokines mentioned above. 

The results obtained need to be interpreted considering some limitations. First, this study has a relatively small sample size, so further replication studies are needed. Second, different CVDs were considered together because of the small sample size. Further studies should identify the effect of individual CVDs on the level of growth factors in SLE patients. Third, there was no group of patients with CVDs alone without autoimmune pathology.

## 5. Conclusions

This study showed the association of an elevated NRG-1β level with the presence of comorbid CVDs in SLE patients. Therefore, NRG-1β can be considered a biomarker of cardiovascular pathology in SLE patients. The level of another growth factor NGFβ was also elevated in SLE patients but was independent of comorbid CVD. Thus, not only cytokines but also growth factors are altered in SLE patients. Further studies are needed to identify the role of growth factors in the pathophysiology of SLE and associated cardiovascular pathology.

## Figures and Tables

**Figure 1 life-14-01305-f001:**
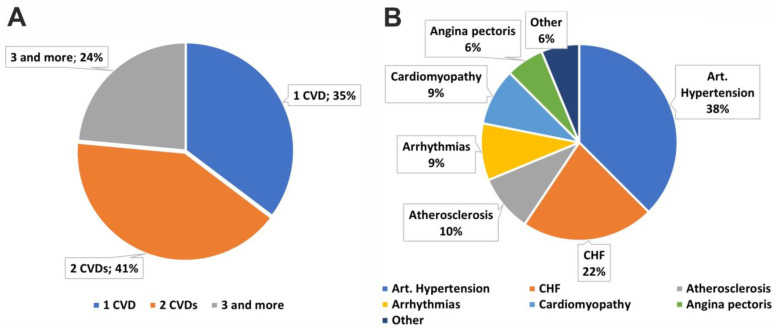
The structure of comorbid CVDs in patients with SLE. (**A**) Proportions of patients with one, two, and three or more CVDs. (**B**) Proportions of specific CVDs among SLE patients.

**Figure 2 life-14-01305-f002:**
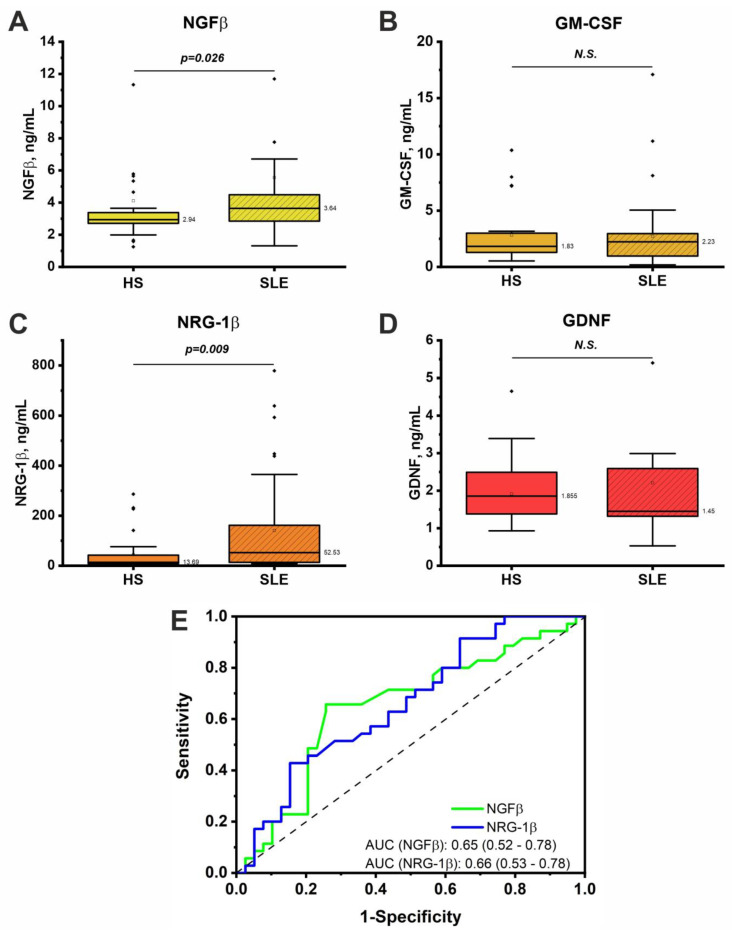
Serum concentration of NGFβ (**A**), GM-CSF (**B**), NRG-1β (**C**), GDNF (**D**) in the total group of SLE patients (n = 35) and healthy individuals (n = 38) determined by Magnetic Luminex assay. The significance of the differences was assessed by the Mann–Whitney test. (**E**) ROC curve reflecting the quality of binary classification of healthy individuals and SLE patients based on NGFβ and NRG-1β levels.

**Figure 3 life-14-01305-f003:**
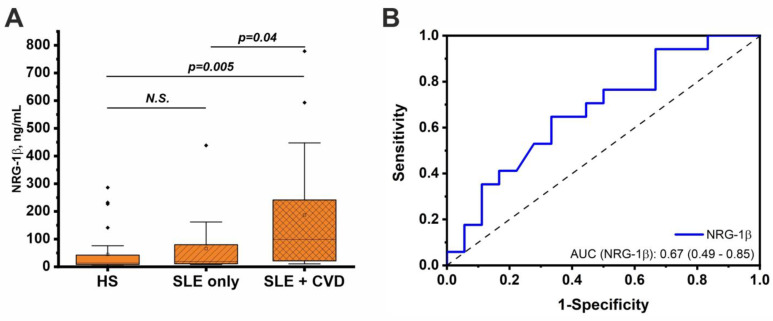
Serum concentrations of NRG-1β in healthy individuals and SLE patients with and without CVDs (**A**). The significance of the differences was assessed using the Kruskal–Wallis test with Dunn’s post hoc test. (**B**) ROC curve reflecting the quality of binary classification of SLE patients with and without CVDs based on NRG-1β level.

**Figure 4 life-14-01305-f004:**
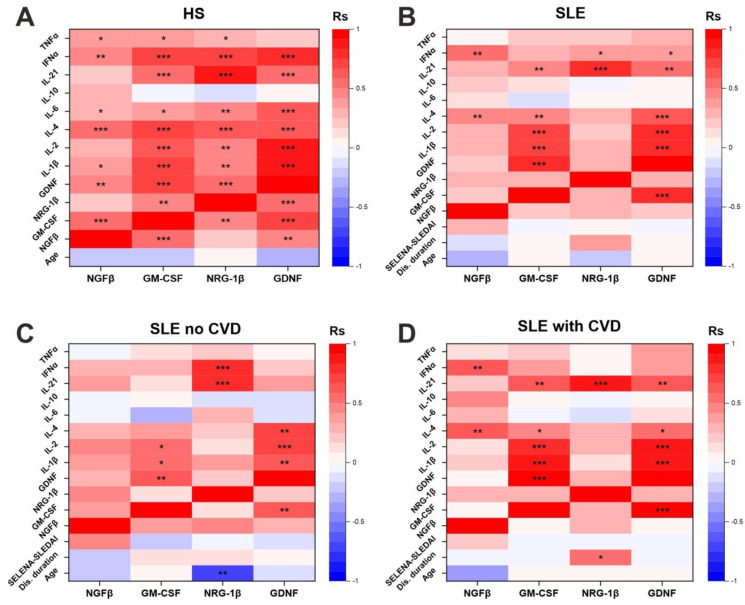
Correlation analysis of serum levels of growth factors with clinical data and cytokine levels in healthy individuals (**A**), the total group of SLE patients (**B**), and in subgroups of SLE patients with (**C**) and without CVDs (**D**). Correlation heatmaps display color-coded Spearman correlation coefficients. Asterisks indicate significant correlations (*—*p* < 0.05, **—*p* < 0.01, ***—*p* < 0.001).

**Figure 5 life-14-01305-f005:**
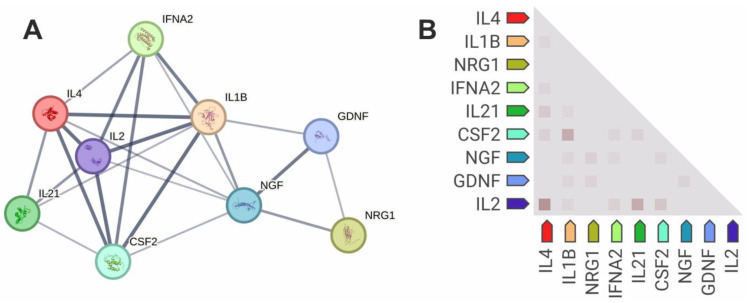
Protein–protein interaction network (**A**) and co-expression heatmap (**B**) of cytokines and growth factors investigated in this study. STRING 12.0 online tool was used for protein–protein interaction and co-expression analysis. Protein–protein interaction network (**A**) reflects known interaction data from curated databases and co-expression analyses. Line thickness indicates the strength of data support. Co-expression heatmap (**B**) reflects data on gene co-expression scores based on RNA expression patterns and on protein co-regulation provided by ProteomeHD. Instead of protein names, gene names are given in the figures (in particular, the CSF2 gene encodes GM-CSF).

**Table 1 life-14-01305-t001:** Clinical characteristics of the study groups.

Parameters	HSs (1)	SLE Total (2)	SLE without CVD (3)	SLE with CVD (4)	Differences
n	38	35	18	17	–
Sex (F/M), %	92.1/7.9	94.3/5.7	94.4/5.6	94.1/5.9	N.S.
Age, years	49 (37–60); 49 ± 12	52 (38–63); 51 ± 14	54 (38–58); 50 ± 13	50 (44–66); 52 ± 15	N.S.
Smokers, %	15.8	17.1	16.7	17.6	N.S.
BMI	21.9 (20.9–24.5)	22.5 (21.8–25.0)	22.4 (21.4–25.0)	22.9 (22.1–25.4)	N.S.
SLE duration, years	–	7 (4–17)	7 (4–15)	11 (4–17)	N.S.
SELENA-SLEDAI score	–	8 (4–10)	8 (5–9)	8 (4–10)	N.S.
SLE phase (active/inactive), %	–	94.3/5.7	88.9/11.1	100/0	N.S.
Anti-dsDNA IgG (ME/mL)	11.92 (8.36–22.02)	64.2 (29.0–190.3)	61.5 (25.9–193.5)	66.8 (37.7–145.8)	1 vs. 2: *p* = 0.0001 1 vs. 3: *p* = 0.0001 1 vs. 4: *p* = 0.0002 3 vs. 4: N.S.
AIP	–	2.78 (2.01–3.62)	2.71 (2.11–3.42)	2.91 (2.49–3.79)	N.S.
Patients received therapy, %	–	100	100	100	–

Note: Data presented as median (Q1–Q3). Age data are also presented as mean ± SD. The significance of the differences was calculated using the Mann–Whitney test. Abbreviations: HSs—healthy subjects, SLE—systemic lupus erythematosus, BMI—Body Mass Index, SELENA-SLEDAI—Safety of Estrogens in Lupus Erythematosus: National Assessment–Systemic Lupus Erythematosus Disease Activity Index, AIP—Atherogenic Index of Plasma.

**Table 2 life-14-01305-t002:** Concentrations of growth factors in serum of SLE patients with and without CVDs compared to healthy subjects.

Growth Factor	HSs (n = 38)	SLE without CVDs (n = 18)	SLE with CVDs (n = 17)
NGFβ	2.9 (1.3–3.4); 4.1 ± 4.1	4 (3–4.5); 4.6 ± 2.3	3.6 (2.9–4.5); 6.7 ± 7.6
GM-CSF	1.8 (0.5–3.0); 2.8 ± 2.6	1.4 (0.9–2.3); 1.8 ± 1.3	2.3 (1.0–3.7); 3.8 ± 4.4
NRG-1β	13.7 (4.4–42); 44 ± 68	19 (9–80); 66 ± 106	99 (22–242); 187 ± 229 *, **
GDNF	1.9 (0.9–2.4); 1.9 ± 0.7	1.4 (1.1–1.9); 1.5 ± 0.7	1.9 (1.3–2.9); 3.0 ± 4.0

Note: Data presented as median (Q1–Q3) and mean ± SD. The significance of the differences was assessed using the Kruskal–Wallis test with Dunn’s post hoc test for multiple comparisons. *—statistically significant differences between SLE patients with CVDs and healthy individuals. **—statistically significant differences between SLE patients with and without CVDs.

## Data Availability

The data presented in this study are available from the corresponding author upon reasonable request.

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
