# Peer review of "Blood Growth Factor Levels in Patients with Systemic Lupus Erythematosus: High Neuregulin-1 Is Associated with Comorbid Cardiovascular Pathology"

_life, 2024, doi:10.3390/life14101305_

Round 1

Reviewer 1 Report

Comments and Suggestions for Authors

The authors discuss the levels of neuregulin-1 in patients with systemic lupus erythematosus (SLE) and its association with cardiovascular disease (CVD).

Abstract: The abstract requires revision. Please remove part about cytokines. Emphasize the aim of the study: the goal of this research was to measure growth factor levels in SLE patients and examine their association with the CVD (minor revision). The authors should also present the main results: NGFβ and NRG-1β concentrations were significantly higher in the total group of 24 SLE patients (n=35) compared to controls (n=38). Please include the exact levels and p-values (minor revision). Similarly, for the statement regarding elevated NRG-1β levels being associated with the presence of CVDs, please provide the exact levels for patients with CVD versus those without CVD (major revision).

Introduction: When discussing CVD, the authors should incorporate key risk factors such as lupus nephritis, nephrotic syndrome (linked to hyperlipidemia), prolonged disease duration, and antiphospholipid syndrome (major revision).

In the early part of the introduction, where the authors mention various mechanisms of SLE and CVD, the sentence about cytokine dysregulation should either be removed or relocated within the introduction: Cytokine dysregulation is thought to play an active role in the pathogenesis of SLE. Thus, cytokines have traditionally been considered potential biomarkers of SLE. The authors should focus more on the parameters measured in this study and their relevance to CVD in SLE (minor revision).

Results: The authors should compare various factors, such as active versus inactive SLE, in the context of CVD. Likewise, lupus nephritis, disease duration, and other variables should be compared (major revision). At the outset, the authors should compare CVD-positive and CVD-negative SLE patients, considering clinical symptoms, ANA profile, antiphospholipid antibodies, medications, and comorbidities, as these factors may influence growth factor levels (major revision).

Discussion and References: These sections are satisfactory as written.

Author Response

Dear Reviewer,

The authors are grateful for the thoughtful analysis of our manuscript and insightful comments. We have carefully considered and applied your feedback to improve our manuscript. We believe that these changes have improved our paper and clarified our data presentation. All revisions were highlighted using the "Track Changes" function in Microsoft Word.

Below we answer your suggestions point by point. Please note that your comments are in italics and our responses are in regular font for readability.

Comment 1: Abstract: The abstract requires revision. Please remove part about cytokines. Emphasize the aim of the study: the goal of this research was to measure growth factor levels in SLE patients and examine their association with the CVD (minor revision). The authors should also present the main results: NGFβ and NRG-1β concentrations were significantly higher in the total group of 24 SLE patients (n=35) compared to controls (n=38). Please include the exact levels and p-values (minor revision). Similarly, for the statement regarding elevated NRG-1β levels being associated with the presence of CVDs, please provide the exact levels for patients with CVD versus those without CVD (major revision).

Response 1: Thank you for this suggestion. We have significantly revised the Abstract. In particular, we added a clearer aim of the study and provided specific values for growth factor concentrations. However, we did not find a way to add p-values as this greatly complicated the presentation and would have required specifying at least 4 different p-values. Instead, we added the term “significantly”, which implies p-values < 0.05. In addition, we left the sentence about cytokines because we consider it important. In this sentence, we emphasize that cytokines are well investigated in SLE, but growth factors are much less well investigated.

Comment 2: Introduction: When discussing CVD, the authors should incorporate key risk factors such as lupus nephritis, nephrotic syndrome (linked to hyperlipidemia), prolonged disease duration, and antiphospholipid syndrome (major revision).

Response 2: Thank you for this valuable suggestion. Indeed, lupus nephritis, nephrotic syndrome, prolonged disease duration, and antiphospholipid syndrome are considered risk factors for cardiovascular disease in SLE. Therefore, we have added these data to the manuscript and supported these statements with relevant literature references (please see third paragraph of the Introduction section).

Comment 3: In the early part of the introduction, where the authors mention various mechanisms of SLE and CVD, the sentence about cytokine dysregulation should either be removed or relocated within the introduction: Cytokine dysregulation is thought to play an active role in the pathogenesis of SLE. Thus, cytokines have traditionally been considered potential biomarkers of SLE. The authors should focus more on the parameters measured in this study and their relevance to CVD in SLE (minor revision).

Response 3: We agree that this part of the Introduction was not sufficiently focused. Therefore, we deleted one sentence about cytokines ((please see fourth paragraph of the Introduction section). But as stated above, in this part we emphasize that cytokines are better studied in SLE than growth factors.

Comment 4: Results: The authors should compare various factors, such as active versus inactive SLE, in the context of CVD. Likewise, lupus nephritis, disease duration, and other variables should be compared (major revision). At the outset, the authors should compare CVD-positive and CVD-negative SLE patients, considering clinical symptoms, ANA profile, antiphospholipid antibodies, medications, and comorbidities, as these factors may influence growth factor levels (major revision).

Response 4: Thank you for this suggestion. Information on SLE patients according to the presence of CVD has already been provided in the manuscript (please see Table 1). Specifically, patients with CVD did not differ in age, sex ratio, proportion of smokers, body mass index, atherogenic index of plasma, and anti-DNA antibody levels compared with patients without CVD. SLE patients with comorbid CVD had slightly longer disease duration, although there were no significant differences compared to patients without CVD. As for patients with lupus nephritis, such patients were not included in the study (information on exclusion criteria is given in section 2.1).

However, we did not aim to compare clinical parameters of patients with and without CVD, so only limited patient information was collected. We are now unable to obtain additional information. For example, we did not collect information on prevalent clinical symptoms, ANA profile, and antiphospholipid antibodies. Although the diagnosis was certainly based on that information. Nevertheless, we have added information on pharmacotherapy for patients with and without CVDs (please see Supplementary Table S2). The percentage of patients taking a specific immunosuppressive drug did not differ between patients with and without CVD. Dosages were also not significantly different. For example, the median dosage of prednisolone was 15 mg/day in the group without CVD and 15 mg/day in the group with CVD. However, the majority of patients without CVD (56.3%) were taking two immunosuppressive drugs, while 53% of patients with CVD were taking three drugs. We have described patient therapy data in the manuscript (please see section 3.1).

In addition, we added data on comorbid pathology (other than CVD) in the groups of patients with and without CVD (please see Supplementary Table S3). In this table, we included only those pathologies that occurred in more than two patients. Osteoporosis, dorsopathies, neuropathies and chronic tonsillitis (in remission) occurred in patients in both groups. However, there were no significant differences in the frequency of these pathologies in the groups of patients with and without CVD. We have included this information in the manuscript (please see section 3.1). In addition, it is important to note that this study had fairly strict exclusion criteria, so many patients with comorbidities that are also common in SLE were excluded.

Thus, we added all available data to best characterize patients with and without CVD.

Comment 5: Discussion and References: These sections are satisfactory as written.

Response 5: Thank you for your positive evaluation of this part of the manuscript.

Thank you very much for taking the time to review this manuscript.

Best regards

Authors

Reviewer 2 Report

Comments and Suggestions for Authors

I read with interest the paper titled "Blood growth factor levels in patients with systemic lupus erythematosus: high neuregulin-1 is associated with comorbid cardiovascular pathology"

The paper is well written, and I have only few comments to add. 

1. Along the manuscript gender should be sex instead. 

2. Regarding the serum concentrations, a lot of outliers seems to be present. Did you test without the outlier inclusion? Have those outliers any specific characteristic?

3. A suggestion is to also create ROC curve reflecting the binary classification of healthy individuals and SLE patients based on NGFβ and NRG-1β levels, stratified by number of CVDs present. I understand that this is quite difficult here, since small sample size, but quite interesting to know. 

4. Figure 3B - (B) ROC curve reflecting the quality of binary classification of SLE patients with and without CVDs based on NRG-1β level. Should this graph have a curve for those with CVD and another for those without? Please briefly explain. 

5. Figure 4. Those heatmaps are quite difficult to interpret without the values of correlations. Could you please add (in supplementary at least)

6. The paper is very well written, however I miss a more robust conclusion of your findings. Please improve the conclusion ideas based on what your results point out. 

Author Response

Dear Reviewer,

We thank the reviewer for the positive evaluation of our study and valuable suggestions. We have carefully considered and applied your feedback to improve our manuscript. All revisions were highlighted using the "Track Changes" function in Microsoft Word.

Below we answer your suggestions point by point. Please note that your comments are in italics and our responses are in regular font for readability.

Comment 1: Along the manuscript gender should be sex instead.

Response 1: We apologize for the inaccuracy. Of course, we understand the difference between the terms sex and gender. We have replaced gender with sex throughout the text.

Comment 2: Regarding the serum concentrations, a lot of outliers seems to be present. Did you test without the outlier inclusion? Have those outliers any specific characteristic?

Response 2: Thank you for this suggestion. Indeed, the samples in our study contain a certain number of outliers. Outliers are quite common in biological data. Outliers were identified by Tukey's test. For graphical representation, the Origin program presents the data as a boxplot, where the horizontal line represents the median, the box boundaries are the quartiles (Q1 and Q3), and the whiskers are the 1.5 interquartile range boundary. Thus, all values above the interquartile range are represented as outliers. However, all data were used to calculate the significance of differences. In addition, we used non-parametric criteria that are insensitive to the presence of outliers. Therefore, the statistical analysis was performed correctly.

As you suggested, we removed outliers from the sample and again assessed the significance of the differences. The results were unchanged, in particular NRG-1β and NGFβ levels were significantly higher in SLE patients compared to healthy individuals. Only the significance of differences (p-values) changed, in the case of NRG-1β from p=0.009 to p=0.004, in the case of NGFβ from p=0.026 to p=0.004. Differences in GM-CSF and GDNF concentrations also remained insignificant. We have included this information in the manuscript (please see section 3.4).

Indeed, it was of interest whether patients with values of growth factor concentrations identified as outliers have specific disease characteristics. We analyzed this group of patients but, due to the small sample size, we were unable to identify any significant associations. We only observed that SLE patients with CVD with high NRG-1β levels had a longer disease duration (please see Supplementary Table S1). However, this association has already been shown by correlation analysis and described in the manuscript (please see Figure 4D and Section 3.4, second paragraph).

Comment 3: A suggestion is to also create ROC curve reflecting the binary classification of healthy individuals and SLE patients based on NGFβ and NRG-1β levels, stratified by number of CVDs present. I understand that this is quite difficult here, since small sample size, but quite interesting to know.

Response 3: We created ROC curves reflecting the quality of binary classification of healthy individuals and SLE patients with one, two, and three+ comorbid CVDs based on NRG-1β levels (please see attached pdf document). The resulting models showed close AUCs (0.77-0.78). However, the model for healthy individuals vs. SLE patients with three+ comorbid CVDs was statistically insignificant (p=0.07). All three models were based on too limited a sample and had large variation in AUC. Therefore, we considered these data to be premature and did not include these data in the manuscript. In the case of NGFβ, all models constructed were insignificant and had little or no predictive power.

Comment 4: Figure 3B - (B) ROC curve reflecting the quality of binary classification of SLE patients with and without CVDs based on NRG-1β level. Should this graph have a curve for those with CVD and another for those without? Please briefly explain.

Response 4: This graph reflects somewhat different. We tested whether SLE patients could be classified into CVD-positive and CVD-negative based on NRG-1β concentration. Therefore, there should be a single curve in the figure. The AUC was 0.67, so the model has an average predictive power.

Comment 5: Figure 4. Those heatmaps are quite difficult to interpret without the values of correlations. Could you please add (in supplementary at least)

Response 5: Indeed, heatmaps can be difficult to follow. We used heatmaps to present the correlation data in a compact way. Following your suggestion, we have added Supplementary Table S8, which presents the correlation analysis results (Spearman correlation coefficients and p-values).

Comment 6: The paper is very well written, however I miss a more robust conclusion of your findings. Please improve the conclusion ideas based on what your results point out.

Response 6: Thank you for this suggestion. We have rewritten the Conclusion to emphasize the most significant findings of this study and future research directions.

Thank you for your thorough analysis of our manuscript and constructive suggestions.

Best regards

Authors

Round 2

Reviewer 1 Report

Comments and Suggestions for Authors

Very good work, congratulations.